# ComAlign: Compositional Alignment in Vision-Language Models

## Abstract

Vision-language models (VLMs) like CLIP have showcased a remarkable ability to extract transferable features for downstream tasks. Nonetheless, the training process of these models is usually based on a coarse-grained contrastive loss between the global embedding of images and texts which may lose the compositional structure of these modalities. Many recent studies have shown VLMs lack compositional understandings like attribute binding and identifying object relationships. Although some recent methods have tried to achieve finer-level alignments, they either are not based on extracting meaningful components of proper granularity or don't properly utilize the modalities' correspondence (especially in image-text pairs with more ingredients). Addressing these limitations, we introduce **Com**positional **Align**ment (**ComAlign**), a fine-grained approach to discover more exact correspondence of text and image components using only the weak supervision in the form of image-text pairs. Our methodology emphasizes that the compositional structure (including entities and relations) extracted from the text modality must also be retained in the image modality. To enforce correspondence of fine-grained concepts in image and text modalities, we train a lightweight network lying on top of existing visual and language encoders using a small dataset. The network is trained to align the entity and relational components across the modalities. Experimental results on various VLMs and datasets demonstrate significant improvements in retrieval and compositional benchmarks, affirming the effectiveness of our plugin model.

## 1 Introduction

Vision-Language Models have achieved impressive results in a broad range of vision-language tasks Tan & Bansal (2019); Bugliarello et al. (2021); Radford et al. (2021); Li et al. (2021a); Zeng et al. (2021); Gan et al. (2022). The popular VLMs like CLIP Radford et al. (2021), and ALIGN Jia et al. (2021) focus on extracting global representation of images and texts by image and text encoders which are trained using a coarse-grained contrastive loss. Recent investigations have revealed that these VLMs struggle to comprehend compositional structures Yuksekgonul et al. (2022); Thrush et al. (2022); Ma et al. (2023), such as binding attributes to the corresponding objects or identifying relationships between subjects and objects.

To provide fine-grained alignment in VLMs, some models, such as PEVL Yao et al. (2022) and X-VLM Zeng et al. (2022), use more supervised datasets. In particular, they require fine-grained supervision, such as bounding box coordinates corresponding to a given entity. On the other hand, VLMs like FILIP Yao et al. (2021) don't need more supervision than image-text pairs and match each fine-grained component to its counterpart precisely. In these models, Fine-grained similarities between image regions and text words are extracted and matched in an unsupervised manner to compute the overall similarity, which is then utilized in standard contrastive learning. PyramidCLIP Gao et al. (2022) aligns image regions and object boxes with descriptive text. This model considers the local and global views for both the image and the text modalities and utilizes both Peer-level and Cross-level Alignment to tackle the mismatch of these modalities.

Despite introducing several fine-grained VLMs, these models don't properly utilize the correspondence of image and text modalities. For example, FILIP Yao et al. (2021) proposes a simple way to create fine-grained

supervision by dividing an image into patches and the descriptive text into tokens. This method considers each word of the text and each patch of the image as an independent component. For example, considering the phrase "A red flower", the "red" and "flower" tokens can be mistakenly matched to disjoint sets of patches without any losses.

To capture the correspondence of the text and image, the meaningful components of these modalities must be extracted. In the textual modality, the Entity Relationship (ER) is utilized as a high-level conceptual model. An entity is a word indicating an object, such as "flower", and phrases like "red flower," which describes both the object and its attribute. Relations such as "a man riding a horse" correspond to a triplet that contains two entities (i.e., subject and object) and the specified relation between them. To provide a basis for better alignment of text and image, we also extract components of similar granularity for the visual modality by considering object-bounding boxes as candidate regions for visual entities and boxes including a pair of object-bounding boxes as candidate regions for visual relations Johnson et al. (2015). Since entities and their attributes appear in the same area of an image in the visual modality, we consider both entities and described entities (with their attributes) as textual entity components. Therefore, the phrase "a red flower" as a textual entity must be matched with the specific region of the image containing a red flower, even if the image also includes flowers of other colors. The VLM can then be trained to match the compositional structures of the two modalities.

In this paper, we propose a method that efficiently utilizes a base VLM and provides a fine-grained VLM. Our method assumes that VLMs like CLIP can extract initial representations for entities of the text and objects of the image and need to be empowered by a lightweight model that can align the structure of the visual and textual modalities. Therefore, after extracting entities and relations from the text and identifying candidate regions for entities and relations from the image, the initial representation of these components is obtained using coarse-grained VLMs like CLIP. To capture the compositional structure, ComAlign is trained on top of the frozen image and text encoders to provide fine-grained alignment of the image and text components. This is done by modeling the compositional structures of the modalities as entity and relational components and using a fine-grained matching strategy. This approach significantly improves zero-shot retrieval and compositional benchmark performance of base models while using a lightweight network and minimal training data. For example, it improves the I2T retrieval performance of CLIP-ViT-B32 on MSCOCO Lin et al. (2014) by 5.60% and T2I by 6.27%, surpassing PyramidCLIP which uses the same backbone while it trains the entire base model from scratch using a large dataset (despite our method only requires training a lightweight network on top of the base model using a much more smaller dataset).

The primary contributions of our work are outlined as follows:

1. We developed a straightforward and efficient preprocessing pipeline that extracts relational and entity components from both image and text modalities directly from raw data, eliminating the need for additional labeling for either modality.

2. We implemented a streamlined approach for unsupervised component matching between text and image modalities, utilizing the components extracted during the preprocessing phase. Additionally, we trained a lightweight network as an extension to the base VLMs, enhancing their compositional understanding and alignment capabilities.

3. We improved performance across compositional benchmarks and various vision-language tasks, including retrieval, by leveraging minimal compositional unlabeled data and eliminating the necessity for retraining the entire VLM.

## 2   Related Works

### 2.1   Vision-Language Models

Vision-language pretraining aims to establish a unified embedding space that bridges vision and language modalities by leveraging large-scale image-text datasets. A prominent approach in this domain involves using contrastive objectives to align image and text embeddings, bringing embeddings of matching pairs closer while pushing unrelated pairs apart.

One of the most influential models in this area is CLIP Radford et al. (2021), which leverages large-scale image-text pairs and has attracted significant attention due to its exceptional performance in retrieval tasks and zero-shot transfer capabilities. Building upon the success of CLIP, models like ALIGN Jia et al. (2021) extended this approach by scaling up the pretraining dataset with noisy image-text pairs to further enhance performance.

Additionally, several studies Li et al. (2021b); Mu et al. (2022); Wu et al. (2021); Cui et al. (2022) have explored methodologies to improve the efficacy and data efficiency of vision-language pretraining. For instance, DeCLIP Li et al. (2021b) and SLIP Mu et al. (2022) propose enhancing model performance by incorporating self-supervised learning techniques. ZeroVL Cui et al. (2022) leverages a lower amount of data by efficient sampling and augments the training data effectively using a novel mix-up method. Furthermore, OTTER Wu et al. (2021) uses a soft image-text matching method for labeling in contrastive learning, which significantly reduces the amount of training data required.

Some methods aim to retain the primary representations of Vision-Language Models while adapting them to downstream tasks using prompt-tuning techniques Zhou et al. (2022); Sun et al. (2022); Bulat & Tzimiropoulos (2022); Guo et al. (2023), which are more parameter-efficient. For instance, CoOp Zhou et al. (2022) employs a set of learnable vectors to optimize context by minimizing classification errors. To handle multi-label settings, where single-label matching methods like CoOp are insufficient, DualCoOp Sun et al. (2022) introduces a pair of differentiable prompts to provide both positive and negative contexts for the target class. Addressing the challenge of overfitting in soft prompt learning, LASP Bulat & Tzimiropoulos (2022) proposes a Language-Aware Soft Prompting approach incorporating hand-crafted textual prompts to enhance robustness. Similarly, for multi-label classification, TaI-DP Guo et al. (2023) employs double-grained prompt tuning to capture both coarse-grained and fine-grained embeddings. In contrast to text-based prompt tuning, some methods explore visual prompt tuning Bahng et al. (2022); Rong et al. (2023); Jia et al. (2022), integrating the input image with learned visual prompts to adapt models effectively for downstream tasks. For Example Bahng et al. (2022), integrates a learnable image perturbation with the image for adaptation to a new task.

However, despite their strengths in achieving remarkable performance, these methods have difficulties understanding the distinct components within each modality and their relationships. In contrast, our proposed approach improves the model's ability to understand these components and their relationships by incorporating a parameter-efficient adaptation technique.

## 2.2 Fine-grained Semantic Matching

Aligning only coarse-grained embeddings across two modalities may overlook the nuanced alignment of components and their interrelationships across two modalities, resulting in an imprecise correspondence. For solving these alignment problems some methods have introduced image-text contrastive learning across various semantic levels Zeng et al. (2022); Yao et al. (2021); Pan et al. (2023); Zhang et al. (2022); Gao et al. (2022). For example, MVPTR Li et al. (2022) creates two levels of semantic components for vision and language. In the visual modality, it uses object boxes with position-aware features and object tags, while in the linguistic modality, it processes text tokens and phrase-level inputs derived from a scene graph. On the other hand, X-VLM Zeng et al. (2022) identifies visual concepts based on associated texts and aligns them with the visual concepts at various levels of granularity. However, these models require fine-grained supervision for matching components during the training process.

On the other hand, various fine-grained image-text matching methods Pan et al. (2023); Zhang et al. (2022); Yao et al. (2021); Gao et al. (2022) have been proposed to enhance matching accuracy between images and textual components. CHAN Pan et al. (2023) uses hard matching, presuming that each textual entity corresponds to a specific image region, employing max pooling over image regions for similarity computation. Additionally, fine-grained VLMs like FILIP Yao et al. (2021) and PyramidCLIP Gao et al. (2022) utilize the fine-grained image-text matching ideas to improve aligning the image and text modalities. FILIP Yao et al. (2021) adopts a bidirectional approach, breaking images into patches and text into tokens, and assumes a correspondence between each text token and image patch using average pooling. PyramidCLIP Gao et al. (2022), in contrast, employs hierarchical semantic matching by integrating local and global views of

both modalities and utilizing peer-level and cross-level matching to address modality mismatches. Despite advancements in image-text matching methods, they fail to break down components into distinct semantic categories while simultaneously matching each component with its counterpart in the other modality, as achieved by our method. Our method extracts three levels of semantic components within each modality and ensures precise matching across corresponding levels, enabling more accurate and meaningful cross-modal alignments via only training a lightweight network on the top of the base VLMs.

## 3 Proposed Method

In this section, we explain our proposed method for extracting and aligning the compositional structure of image and text. Initially, we extract fine-grained components from images and texts. These components, along with the entire image and text, are processed by a frozen pre-trained VLM to obtain representations. We then feed them into our ComAlign encoders to capture the interactions between the fine-grained and coarse-grained features within each modality. By aligning corresponding concepts across modalities, we achieve representations that effectively capture both fine-grained and coarse-grained information.

### 3.1 Preprocessing: Extracting Fine-Grained Components

The structured nature of text enables us to extract **entity** and **relational components** more accurately compared to images, which lack an inherent structure. To address this limitation in images, we employ a pre-trained object detector that identifies candidate entities and relationships. This approach introduces a preprocessing pipeline designed to extract **fine-grained components** from both text and image data that initially lack detailed annotations. Our method focuses on two distinct types of fine-grained components:

1. **Entity Components:** These represent individual objects in images or entities in text (along with their corresponding adjective, if available).

2. **Relational Components:** These capture regions showcasing interactions between two objects in images or parts of texts showing relationships between two entities.

#### 3.1.1 Textual Modality Preprocessing

For textual data, we utilize SpaCy's *en_core_web_sm* pre-trained English language model Honnibal et al. (2020). This model extracts:

- **Textual Entity Components:** Represented as pairs $e = (\text{adjective}, \text{entity})$, where nouns serve as entities, and adjectives provide additional context. In cases where no adjective is associated with an entity in the text, the entity component can be represented as $e = (\text{entity})$.

- **Textual Relational Components:** Represented as triplets $r = (\text{entity}_1, \text{relation}, \text{entity}_2)$, where *relation* describes either spatial or non-spatial relationship between two entities identified within the text.

#### 3.1.2 Visual Modality Preprocessing

For images, we leverage a pre-trained object detector to identify:

- **Visual Entity Components:** Object-bounding boxes serve as entity components, each scored by the confidence of the object detector $(p_i)$, which reflects the likelihood of its *objectness*. The top-$N$ objects with the highest scores are retained as the final set of visual entity components.

- **Visual Relational Components:** Candidate visual relational components are generated by considering all possible pairs of object-bounding boxes from the extracted visual entity components. For each pair, we define a minimal bounding box that encapsulates both objects. The confidence score of a visual relation component is calculated as the product of the scores of the two entities

$(p_i \times p_j)$. The top-$M$ visual relational components with the highest scores are selected as the final set.

This dual-modality preprocessing pipeline ensures that both textual and visual components are represented at a fine-grained level, facilitating further downstream analysis and integration across modalities.

## 3.2 Architecture

First, we embed the extracted textual and visual components. More precisely, the object and relation bounding boxes in the image are cropped, resized, and then embedded by the image encoder of the base VLM. Textual entities and relations are also embedded by the text encoder of the base VLM. Moreover, the frozen VLM also embeds the whole image and text. The obtained representations for the $N$ entity components, the $M$ relational components, and the global representations of image $i$ are shown as $\{h_i^{I,e}\}_{e=1}^N$, $\{h_i^{I,r}\}_{r=1}^M$, and $h_i^{I,g}$, respectively. The corresponding representations for the entities, relations, and the whole input for text $j$ are also denoted as $\{h_j^{T,e}\}_{e=1}^N$, $\{h_j^{T,r}\}_{r=1}^M$, and $h_j^{T,g}$, respectively. $N$ and $M$ are treated as hyper-parameters, determined before training. The number of extracted entities and relationships is adjusted by truncating excess components or padding to reach the determined numbers $N$ and $M$, respectively.

We want to improve the representations of fine-grained components since they have been extracted individually by the base VLM. To this end, we employ a simple two-layer transformer architecture to find the contextualized representations of components that also have been enforced to consider the fine-grained and coarse-grained correspondence of the image and text modality. Therefore, the representation of image $i$ and text $j$ are fed as $h_i^I = [h_i^{I,g}, h_i^{I,e_1}, ..., h_i^{I,e_N}, h_i^{I,r_1}, ..., h_i^{I,r_M}]$ and $h_j^T = [h_j^{T,g}, h_j^{T,e_1}, ..., h_j^{T,e_N}, h_j^{T,r_1}, ..., h_j^{T,r_M}]$ to the ComAlign image and text encoder, respectively. Specifically, the contextualized representations are obtained as:

$$z_i^I = F_{\theta_I}(h_i^I), \qquad z_j^T = G_{\theta_T}(h_j^T), \tag{1}$$

where the ComAlign encoder networks $F_{\theta_I}$ and $G_{\theta_T}$ are two-layer transformer models for improving vision and language representations, respectively.

## 3.3 Training Objectives

The goal is to ensure that each image's representation closely aligns with its corresponding text while simultaneously differing significantly from the representations of unrelated texts. To achieve this, we must match the corresponding components in the image and text pair. First, we define the fine-grained matching method for aligning image and text representations. Then, we indicate how entity, relational, and global similarity between image and text representations are obtained.

**Fine-Grained Matching:** We intend to match the corresponding components of two modalities. To do this, we use the matching strategy introduced in FILIP Yao et al. (2021) and align the two set of representation vectors $\{x_k\}_{k=1}^C$ and $\{x_l'\}_{l=1}^{C'}$, using the following Fine-Grained-Matching (FGM) function:

$$FGM(\{x_k\}_{k=1}^C, \{x_l'\}_{l=1}^{C'}) = \text{mean}_{1 \le k \le C} \left\{ \max_{1 \le l \le C'} \left\{ x_k^T x_l' \right\} \right\}. \tag{2}$$

in which for each vector in the first set, the most similar vector from the second set is identified based on dot product similarity, treating these two vectors as the matched pair, and then compute the average similarity across all these matched vector pairs.

**Entity and Relational Components Similarity:** We compute the entity-based similarity between images and text by defined fine-grained matching. Image-to-Text (I2T) and Text-to-Image (T2I) similarities of image $i$ and text $j$ is defined as follows:

$$s_{i,j}^{I2T,E} = FGM(\{z_i^{I,e}\}_{e=1}^N, \{z_j^{T,e}\}_{e=1}^N), \qquad s_{i,j}^{T2I,E} = FGM(\{z_j^{T,e}\}_{e=1}^N, \{z_i^{I,e}\}_{e=1}^N), \tag{3}$$

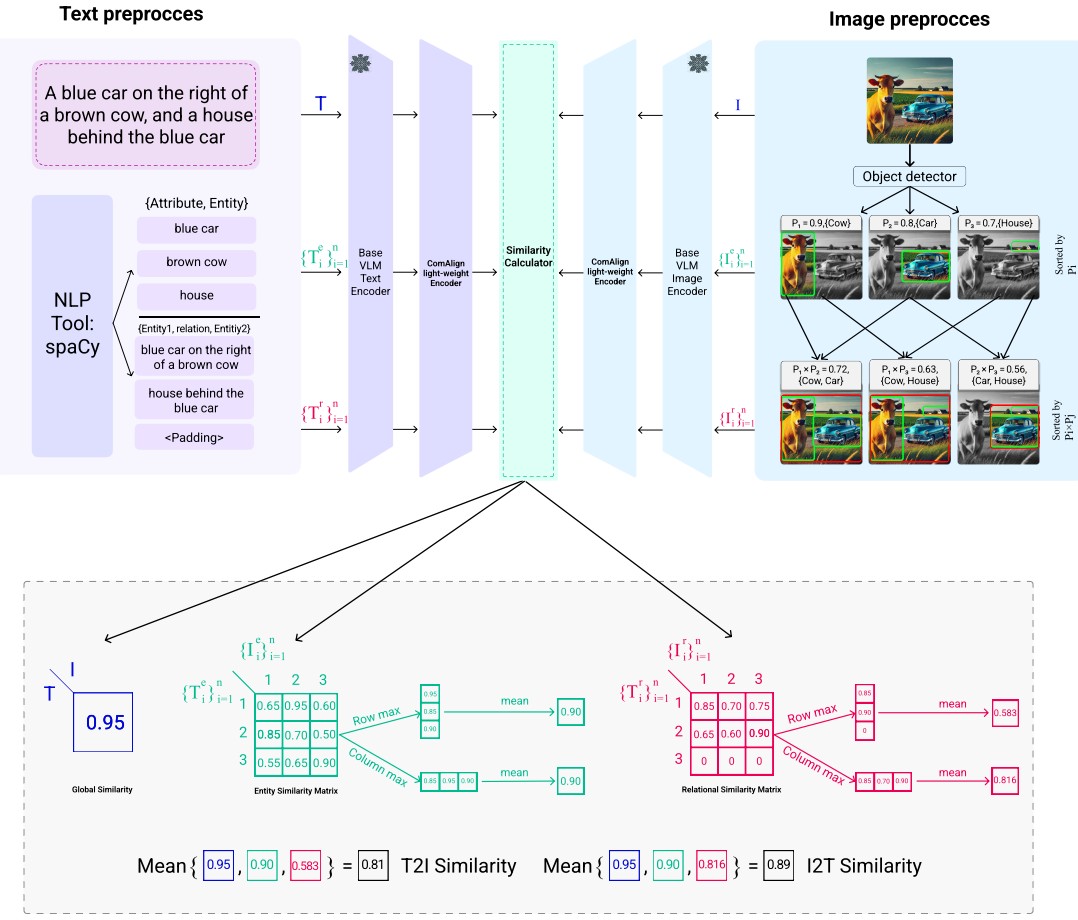

Figure 1: Overview of the proposed method. Given a batch of image-text pairs, each image and text is pre-processed by object-detector and NLP tools to extract entity and relational components. These components, along with the original image and text, are processed by a base VLM to obtain visual and textual representations. These are then passed through our ComAlign image and text encoders. We calculate the similarity score between an image and a text using three types of scores: 1) Coarse-grained similarity: Calculated as the dot product of the global features of the image and text. 2) Fine-grained entity-based similarities: The entity similarity matrix is obtained by calculating the cosine similarity between each pair of the visual entity component representation and the textual entity component representation. 3) Fine-grained relation-based similarities: Similarly, the relation similarity matrix is computed according to the cosine similarity of all pairs of visual and textual relation representations. By employing Fine-Grained Matching on the obtained matrices, the whole entity-based similarity and relation-based similarity between the image and text are found (for both Text2Image and Image2Text directions). The final aggregated similarities are used during both the contrastive training and inference process.

where $z_i^{I,e} \in \mathbb{R}^D$ and $z_j^{T,e} \in \mathbb{R}^D$ shows the representation of the entity component $e$ of image $i$ and text $j$ respectively, and $N$ denotes the number of entity components.

Relational components are matched similarly:

$$s_{i,j}^{I2T,R} = FGM(\{z_i^{I,r}\}_{r=1}^M, \{z_j^{T,r}\}_{r=1}^M), \qquad s_{i,j}^{T2I,R} = FGM(\{z_j^{T,r}\}_{r=1}^M, \{z_i^{I,r}\}_{r=1}^M). \tag{4}$$

where $z_i^{I,r} \in \mathbb{R}^D$ and $z_j^{T,r} \in \mathbb{R}^D$ shows the representation of the relational component $r$ of image $i$ and text $j$ respectively, and $M$ denotes the number of relational components.

**Global Similarity** We use the standard dot product for computing the similarity between two global features, considering $z_i^{I,g} \in \mathbb{R}^D$ and $z_i^{T,g} \in \mathbb{R}^D$:

$$s_{i,j}^{I2T,G} = s_{i,j}^{T2I,G} = (z_i^{I,g})^T z_j^{T,g}. \tag{5}$$

The loss function is the sum of the contrastive losses for each of the entity, relational, and global features, with the similarity calculated differently for each category. Specifically, the image-to-text and text-to-image contrastive losses are defined as:

$$\mathcal{L}_i^{I2T} = f_i(\{s_{i,j}^{I2T,E}\}_{j=1}^B) + f_i(\{s_{i,j}^{I2T,R}\}_{j=1}^B) + f_i(\{s_{i,j}^{I2T,G}\}_{j=1}^B), \tag{6}$$

$$\mathcal{L}_i^{T2I} = f_i(\{s_{i,j}^{T2I,E}\}_{j=1}^B) + f_i(\{s_{i,j}^{T2I,R}\}_{j=1}^B) + f_i(\{s_{i,j}^{T2I,G}\}_{j=1}^B), \tag{7}$$

where $f_i$ is defined as:

$$f_i(\{s_{i,j}\}_{j=1}^B) = -\log \frac{\exp(s_{i,i})}{\sum_{j=1}^B \exp(s_{i,j})}. \tag{8}$$

The final loss in a batch is computed by mean of I2T and T2I losses as:

$$\mathcal{L} = \frac{1}{2B} \sum_{i=1}^B (\mathcal{L}_i^{I2T} + \mathcal{L}_i^{T2I}). \tag{9}$$

Figure 2 shows an example of this calculation process.

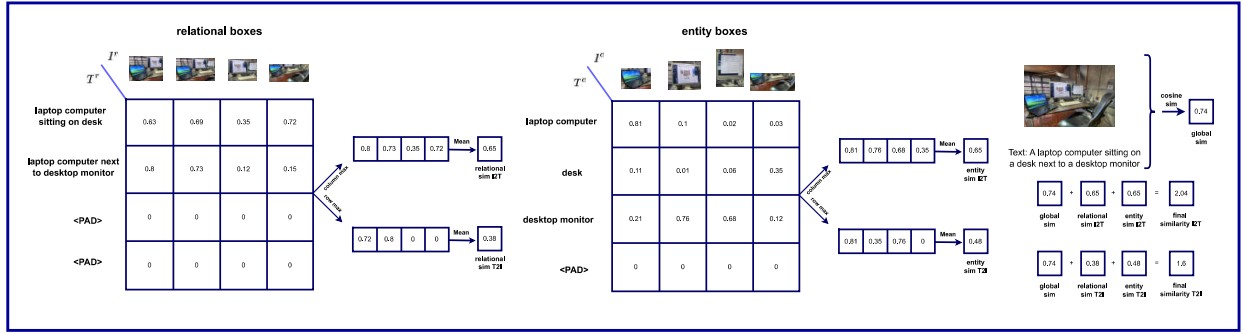

Figure 2: Illustration of the process of calculating Image-to-Text (I2T) and Text-to-Image (T2I) similarity, including global, entity, and relational components.

### 3.4 Inference

During inference, the fine-grained and coarse-grained representation of the images and texts $(z^I, z^T)$ are obtained according to the proposed method in Section 3.2. To calculate the T2I and I2T similarities between each image-text pair, we consider a weighted sum of corresponding fine and coarse-grained similarities $(s^{\cdot,E}, s^{\cdot,R}, s^{\cdot,G})$ along with the dot product of the base VLM image and text representations $(h^{I,g}, h^{T,g})$.

Additionally, we use fine-grained T2I similarities to calculate the final I2T similarity score. This approach is based on the assumption that not all visual components of an image are captured in its caption, as it often fails to describe every detail present in the image. In fact, the expectation of finding a corresponding matched textual component for each visual component is somewhat unrealistic. Therefore, incorporating T2I fine-grained similarities could help compensate for this weakness. The final similarity score is formulated as follows:

$$s_{i,j}^{I2T} = (h_i^{I,g})^T h_j^{T,g} + \alpha_1(s_{i,j}^{I2T,G} + s_{i,j}^{I2T,E} + s_{i,j}^{I2T,R}) + \alpha_2(s_{i,j}^{T2I,E} + s_{i,j}^{T2I,R}), \tag{10}$$

$$s_{i,j}^{T2I} = (h_i^{I,g})^T h_j^{T,g} + \beta_1(s_{i,j}^{T2I,G} + s_{i,j}^{T2I,E} + s_{i,j}^{T2I,R}). \tag{11}$$

Here, $s_{i,j}^{I2T/T2I,E/R/G}$ are calculated according to Equations 3, 4, and 5. Also, $\alpha_1$, $\alpha_2$ and $\beta_1$ are considered as hyper-parameters.

## 4 Experiments

### 4.1 Experimental Setup

**Base VLMs** We applied our alignment method to two CLIP model backbones released by OpenAI: ViT-B/32 and ViT-L/14. Furthermore, we tested our method on four other models: NegClip Yuksekgonul et al. (2022), CoCa Yu et al. (2022), Vitamin Chen et al. (2024), and Mobileclip Vasu et al. (2024). NegCLIP enhances contrastive learning by utilizing negative samples to better distinguish between similar images and texts. CoCa combines a caption generation objective with contrastive learning to improve fine-grained understanding. ViTamin introduces a scalable vision model with a three-stage architecture that integrates two MBConv block stages and a final Transformer block stage, combining the advantages of convolutional and transformer-based methods to maintain high feature resolution and scalability. MobileCLIP optimizes lightweight image-text models through multi-modal reinforced training, leveraging knowledge transfer from robust CLIP encoders and an image captioning model via a reinforced dataset.

We used the NegCLIP checkpoint from its official GitHub repository[1], as well as the CoCa model, which was accessed via an OpenCLIP checkpoint Ilharco et al. (2021) which was trained on the LAION-2b dataset Schuhmann et al. (2022). Both models are built on the ViT-B/32 network backbone. Additionally, for ViTamin-B-LTT and MobileCLIP-B, we used OpenCLIP checkpoints, with each model pre-trained on different versions of the DataComp dataset Gadre et al. (2024).

**Implementation Details** The training process was performed on an Nvidia 1080 GPU, with each base model training completed within 4 hours (All our experiments were also performed on the specified GPU card). This minimal training time and low GPU VRAM requirement are due to our lightweight network, which consists of a two-layer transformer and a relatively small training dataset of approximately 100,000 image-text pairs. All models were trained using the AdamW optimizer Loshchilov & Hutter (2017) and a StepLR learning rate scheduler, with a batch size of 1600 image-text pairs. Positional encoding was implemented in the transformer layers as described in Vaswani et al. (2017). We set the maximum number of both entity and relational components, $N$ and $M$, to 10. We used spaCy to extract components from the text, and YOLOv9 Wang et al. (2024) as the object detector to identify visual components. The training process is depicted in Algorithm 1.

---

[1] https://github.com/mertyg/vision-language-models-are-bows

We performed hyper-parameter tuning for training and inference using a subset of the training dataset we composed. Further details can be found in Appendix A.2.

---

**Algorithm 1** Training process of ComAlign

---

1: Initialize preprocessed dataset $\mathcal{D}$ and batch size $\mathcal{B}$
2: Initialize transformer models $F_{\theta_I}$ and $G_{\theta_T}$
3: **for** update step $= 1$ to $M$ **do**
4:     Sample a batch of image-text pairs $(h_i^I, h_i^T)_{i=1}^{\mathcal{B}}$ from $\mathcal{D}$
5:     Compute image and text representations: $z_i^I = F_{\theta_I}(h_i^I)$ and $z_i^T = G_{\theta_T}(h_i^T)$
6:     Decompose $z_i^I$ and $z_i^T$ into entity, relational, and global features $z_i^{I,e}, z_i^{I,r}, z_i^{I,g}$ and $z_i^{T,e}, z_i^{T,r}, z_i^{T,g}$, respectively.
7:     Calculate similarity scores for entities, relations, and global features: $s^{\cdot,E}$, $s^{\cdot,R}$, $s^{\cdot,G}$ using Equations 3, 4, and 5
8:     Compute the final loss using Equations (6) through (9)
9:     Update model parameters $\theta_I$ and $\theta_T$ of $F_{\theta_I}$ and $G_{\theta_T}$ using the Adam optimizer
10: **end for**

---

## 4.2 Datasets

**Visual Genome** This dataset comprises 100,000 images with fine-grained annotations. Each image includes two types of annotations: 1) Attribute Annotations: These annotations describe the objects and their attributes. 2) Relational Annotations: These annotations consist of triplets in the format (Subject, Object, Relation).

**MSCOCO** This dataset contains approximately 100,000 images, each accompanied by five descriptive captions. We used the version of MSCOCO released in 2017.

**Flickr30K** This dataset includes around 30,000 images, each with several captions similar to MSCOCO.

## 4.3 Zero-shot Image-Text Retrieval

Zero-shot image-text retrieval consists of two sub-tasks: image-to-text retrieval and text-to-image retrieval. We use Flickr30K Plummer et al. (2015) and Visual Genome Yuke Zhu (2017) datasets for training. For the Flickr30K zero-shot retrieval, we trained our model on around 100K image-text pairs from the Visual Genome dataset, excluding images that are also part of Flickr30K. For the MSCOCO zero-shot retrieval, we removed images from the Visual Genome dataset present in MSCOCO alongside training data from Flickr30K, resulting in a dataset of approximately 80K image-text pairs.

We evaluate the performance of our alignment method on the Flickr30K and MSCOCO datasets, comparing it to the base VLMs and PyramidClip Gao et al. (2022). PyramidClip is pre-trained from scratch on a dataset containing 143 million samples. It introduces multiple semantic levels and employs contrastive alignment between both peer-level and cross-level components. This approach enhances the model's compositional and fine-grained understanding, resulting in improved retrieval performance.

Table 1 shows our results compared to the baselines. We observe extensive performance improvement for CLIP-ViT-B/32 and CLIP-ViT-L/14 in image-to-text (I2T) and text-to-image (T2I) retrieval on both datasets. Additionally, our method improves CoCa's performance (which also leverages captioning objective that leads to better compositional understanding), except in text-to-image retrieval on the Flickr30K dataset, where our performance was comparable to the base model. Notably, while NegCLIP employs a full-finetuning approach that improves compositional understanding using hard negative image-text pairs, our contribution complements theirs via fine-grained matching of components leveraging only the available weak supervision in the form of image-text pairs. Hence, when applying our method on NegCLIP, we achieved up to a 3.06% improvement in image-to-text retrieval on Flickr30K.

Interestingly, when applied to CLIP-ViT-B/32, COCA-ViT-B/32, and NegClip-ViT-B/32 that have the same encoder backbones, our model outperforms PyramidCLIP in image-to-text retrieval. It is worth to mention

Table 1: Zero-shot image-text retrieval results on MSCOCO and Flickr30K datasets.

| Method | MSCOCO | | | | | | Flickr30K | | | | | |
| | image-to-text | | | text-to-image | | | image-to-text | | | text-to-image | | |
| | R@1 | R@5 | R@10 | R@1 | R@5 | R@10 | R@1 | R@5 | R@10 | R@1 | R@5 | R@10 |
|---|---|---|---|---|---|---|---|---|---|---|---|---|
| CLIP-ViT-B/32 | 50 | 74.96 | 83.28 | 30.35 | 54.77 | 66.09 | 78.59 | 95.36 | 97.63 | 59.72 | 84.83 | 90.67 |
| CLIP-ViT-B/32 + ComAlign | 55.60 | 79.72 | 86.88 | 36.62 | 63.55 | 74.77 | 82.24 | **97.04** | **98.61** | 66.27 | 88.22 | 93.11 |
| Relative gain | 5.60 | 4.76 | 3.60 | 6.27 | 8.78 | 8.68 | 3.65 | 1.68 | 0.98 | 6.55 | 3.39 | 2.44 |
| COCA-ViT-B/32 | 54.04 | 77.72 | 86.08 | 35.89 | 61.20 | 71.97 | 82.64 | 95.36 | 97.63 | 64.31 | 86.96 | 91.77 |
| COCA-ViT-B/32 + ComAlign | 56.42 | 80.30 | 88.06 | 37.29 | 63.98 | 74.93 | 84.22 | 96.64 | 98.32 | 63.07 | 86.31 | 92.05 |
| Relative gain | 2.38 | 2.58 | 1.98 | 1.40 | 2.78 | 2.96 | 1.58 | 1.28 | 0.69 | -1.24 | -0.65 | 0.28 |
| NegClip-ViT-B/32 | 56.84 | 80.72 | 88.06 | 41.56 | 68.68 | 78.92 | 83.03 | 95.56 | 97.53 | 68.73 | 89.90 | 94 |
| NegClip-ViT-B/32 + ComAlign | **58.60** | **82.62** | **89.42** | **42.16** | **69.82** | **79.93** | **86.09** | 96.74 | 98.22 | **69.11** | **90.43** | **94.49** |
| Relative gain | 1.76 | 1.90 | 1.36 | 0.60 | 1.14 | 1.01 | 3.06 | 1.18 | 0.69 | 0.38 | 0.53 | 0.49 |
| PyramidCLIP-ViT-B/32 | 52.6 | 79.04 | 86.8 | 39.64 | 65.14 | 75.37 | 80.96 | 96.64 | **98.61** | 67.31 | 89.30 | 93.53 |
| CLIP-ViT-L/14 | 56.08 | 79.6 | 86.86 | 35.31 | 59.96 | 70.14 | 86.29 | 97.33 | **99.30** | 67.83 | 88.85 | 93.25 |
| CLIP-ViT-L/14 + ComAlign | **61.86** | **84.34** | **90.80** | **42.40** | **69.04** | **78.78** | **89.25** | **97.92** | **99.30** | **73.19** | **91.97** | **95.44** |
| Relative gain | 5.78 | 4.74 | 3.94 | 7.09 | 9.08 | 8.64 | 2.96 | 0.59 | 0 | 5.36 | 3.12 | 2.19 |
| ViTamin-B-LTT | 59.56 | 82.48 | 89.08 | 39.74 | 65.48 | 75.11 | 85.50 | 97.33 | 99.01 | 69.03 | 89.46 | 93.53 |
| ViTamin-B-LTT + ComAlign | **61.26** | **83.16** | **90.02** | **42.77** | **68.73** | **78.10** | **87.27** | **98.12** | **99.30** | **73.74** | **91.89** | **95.24** |
| Relative gain | 1.7 | 0.68 | 0.94 | 3.03 | 3.25 | 2.99 | 1.77 | 0.79 | 0.29 | 4.71 | 2.43 | 1.71 |
| MobileCLIP-B | 69.5 | 88.26 | 93.34 | 49.66 | 74.32 | 82.69 | 91.42 | 98.32 | 99.50 | 78.75 | 93.55 | 96.25 |
| MobileCLIP-B + ComAlign | **70.92** | **89.02** | **93.8** | **51.66** | **76.56** | **84.34** | **92.60** | **98.71** | **99.60** | **80.55** | **94.53** | **97.08** |
| Relative gain | 1.42 | 0.76 | 0.46 | 2 | 2.24 | 1.65 | 1.18 | 0.39 | 0.1 | 1.8 | 0.98 | 0.83 |

that we only train a small network on top of the base models using a small dataset (100K samples) while PyramidCLIP trained the entire network with a massive dataset (143M samples). Our careful construction of entity and relational components, combined with a straightforward matching strategy, enables our method to utilize the fine-grained information in the base models effectively.

Additionally, in Vision-Language Models such as CLIP-ViT-L/14, which employs a larger Vision Transformer with smaller patches that outperforms CLIP-ViT-B/32, our proposed model again achieves notable enhancements, improving performance by 5.78% in image-to-text zero-shot retrieval and 7.09% in text-to-image zero-shot retrieval on the MSCOCO dataset. Moreover, our model is applied to two recent VLMs, ViTamin and MobileCLIP, to enhance their compositional understanding. ViTamin introduces a vision model that combines convolutional and transformer-based approaches to maintain high feature resolution and scalability, achieving superior performance in downstream tasks compared to previously mentioned models. MobileCLIP, on the other hand, leverages reinforced training and knowledge transfer from robust CLIP encoders and an image captioning model and significantly enhances performance in downstream tasks, such as zero-shot image-to-text and text-to-image retrieval. The results in Table 1 demonstrate the effectiveness of incorporating our compositional alignment module on the top of these models in achieving higher image-to-text and text-to-image retrieval performance.

### 4.4 Compositional Benchmarks

We use two benchmarks to evaluate the compositional capabilities of our method. The ARO benchmark Yuksekgonul et al. (2022) is designed to evaluate VLMs' ability to understand various attributes, relationships, and orderings. We utilize two parts of the ARO benchmark: *1) VG-Attribution:* This benchmark involves binary classification tasks where each image is paired with two captions. One caption correctly describes two objects along with their attributes, while the other caption is incorrect because it swaps the attributes of the objects. The models' ability to identify the correct caption is assessed, thereby evaluating their attribute-binding capability. *2) VG-Relation:* Similar to VG-Attribution, this part also consists of binary classification tasks. For each image, there is one correct caption and one incorrect one. The correct caption describes two objects and their relationship, whereas in the incorrect caption, the objects are swapped. This task measures the models' ability to accurately understand relationships and orderings between objects in images.

Table 2: Results on compositional benchmarks, including attribute binding (VG-Att), subject-object binding (VG-Rel), and SVO-Probes.

| Method | VG-Rel | VG-Att | SVO Probes |
|---|---|---|---|
| CLIP-ViT-B/32 | 58.82 | 61.05 | 67.63 |
| CLIP-ViT-B/32 + ComAlign | 61.95 | 66.60 | 70.07 |
| Relative Gain | 3.13 | 5.55 | 2.44 |
| COCA-ViT-B/32 | 42.30 | 57.80 | 72.47 |
| COCA-ViT-B/32 + ComAlign | 63.46 | 61.36 | 72.60 |
| Relative Gain | 21.16 | 3.56 | 0.13 |
| NegClip-ViT-B/32 | 78.61 | 68.98 | 72.41 |
| NegClip-ViT-B/32 + ComAlign | 79.49 | 71.79 | 72.60 |
| Relative Gain | 0.88 | 2.81 | 0.19 |
| CLIP-ViT-L/14 | 60.98 | 60.96 | 70.81 |
| CLIP-ViT-L/14 + ComAlign | 59.53 | 65.90 | 74.01 |
| Relative Gain | -1.45 | 4.94 | 3.2 |
| ViTamin-B-LTT | 38.59 | 55.06 | 72.37 |
| ViTamin-B-LTT + ComAlign | 41.41 | 63.23 | 72.79 |
| Relative Gain | 2.82 | 8.17 | 0.42 |
| MobileCLIP-B | 53.67 | 65.78 | 73.84 |
| MobileCLIP-B + ComAlign | 55.54 | 70.21 | 74.12 |
| Relative Gain | 1.87 | 4.43 | 0.28 |

SVO-Probes Hendricks & Nematzadeh (2021) is another benchmark designed to evaluate VLMs' understanding of relationships and attributes. The benchmark comprises a dataset of paired images labeled as positive or negative, accompanied by a positive caption and a positive and negative triplet. Each positive caption contains the subject, verb, and object present in its positive triplet, while each negative triplet differs in one of these three parts from the positive triplet. To create a negative caption, we replace positive triplets in the caption with their negative counterpart, which enables the assessment of the model's understanding in both entity recognition (subject, object replacement) and relational understanding (verb replacement) by matching the images with their corresponding positive or negative captions in a binary retrieval task.

We applied our method to CLIP-ViT-B/32, CLIP-ViT-L/14, CoCa, NegCLIP, ViTamin-B-LTT, and MobileCLIP-B and evaluated their performance on the specified compositional benchmarks. As shown in Table 2, we observe general performance improvements, with only a few exceptions. These enhancements are attributed to the fine-grained components we constructed during training. The entity components enhance the models' ability to bind objects with their attributes, while the relational components improve their understanding of relationships. Notably, the improvement gap is typically higher in the VG-Attribution benchmark compared to the VG-Relation benchmark. This difference may be because VG-Relation also assesses the models' capability to recognize order, which is not addressed by our method.

### 4.5 Zero-shot Classification

We evaluated our proposed method on five different zero-shot classification datasets across multiple base VLMs, with the results presented in Appendix A.1. As demonstrated by the results, our method enhances the performance of base VLMs in compositional understanding and related tasks. Importantly, it does not negatively impact or reduce the performance of these models on other downstream tasks, such as zero-shot classification.

### 4.6 Ablation Study

We conducted several experiments to evaluate our model's performance under different hyper-parameters and ablation conditions. All experiments used CLIP-ViT-B/32 as the base model, trained exclusively on the Visual Genome and Flickr datasets. The results are reported for the image-to-text and text-to-image retrieval tasks on the MSCOCO validation split. Furthermore, only the image transformer was trained.

**Loss Term Study** In this experiment, we examine the effect of fine-grained entity, relation, and global similarities by removing them from the final loss calculation. In this part, we only prevent the addition of

Table 3: Ablation study of each loss term (Removal of just loss term) and feature (Removal of loss term alongside exclusion from the transformer) on MSCOCO Zero-Shot Retrieval.

| Base VLM | Loss Term | | | MSCOCO (Loss Term) | | Feature | | | MSCOCO (Feature) | |
|---|---|---|---|---|---|---|---|---|---|---|
| | Global | Entity | Relation | I2T R@1 | T2I R@1 | Global | Entity | Relation | I2T R@1 | T2I R@1 |
| | ✓ | ✓ | ✓ | 54.60 | 37.07 | ✓ | ✓ | ✓ | 54.60 | 37.07 |
| | ✓ | ✓ | | 52.24 | 36.50 | ✓ | ✓ | | 53.24 | 35.83 |
| ViT-B/32 | ✓ | | ✓ | 52.76 | 35.19 | ✓ | | ✓ | 52.64 | 35.61 |
| | ✓ | | | 53.76 | 36.25 | ✓ | | | 52.10 | 34.14 |
| | | ✓ | ✓ | 52.26 | 36.37 | | ✓ | ✓ | 52.38 | 35.16 |

the similarity term of these parts to the final loss in Eqs. 6 and 7 while still allowing all three features to attend to each other within the transformer architecture. Additionally, the omitted similarity terms will not be used during inference. The results can be seen in Table 3.

In the second part of the experiment, in addition to excluding the unchecked terms from the loss function, we prevent the omitted features from interacting with others within the transformer. By doing so, as seen in Table 3, we generally observe a further decrease in performance, compared to only removing the loss term. This indicates that, although fine-grained features are not utilized during inference, their interaction with global features improves overall alignment. It also highlights the effectiveness of the extracted fine-grained components and our fine-grained matching methodology.

**Network Architecture** In this section, we examine the effects of different encoder architectures as alternatives to transformer layers. We experimented with two new architectures: one using fully connected layers that process both coarse and fine-grained features through a single network and another using two distinct networks for each feature type. As shown in Table 4, the transformer-based architecture yields superior results, likely due to its ability to facilitate interactions between different features.

Table 4: Ablation study of Architecture and Number of Transformer Layers on MSCOCO Zero-Shot Retrieval.

| Architecture | MSCOCO (Architecture) | | Number of Layers | MSCOCO (Layers) | |
|---|---|---|---|---|---|
| | I2T R@1 | T2I R@1 | | I2T R@1 | T2I R@1 |
| Transformer | 54.78 | 37.60 | 1 | 54.78 | 37.60 |
| FC (shared network) | 53.62 | 35.33 | 2 | 54.60 | 37.07 |
| FC (separate networks) | 53.62 | 34.87 | 4 | 28.94 | 19.71 |

**Network Layers** In this experiment, we examined how the size of the appended network affects model performance. Specifically, we increased the number of layers in the transformer network to enhance its expressive power. However, as shown in Table 4, increasing the number of layers to four significantly decreased performance. We believe this is due to the small size of our dataset, which leads to overfitting when using a larger network.

## 4.7 Visualization

As illustrated in the similarity matrices of Figure 3, our alignment surpasses CLIP in matching textual and visual components for both entity and relation. We compute the similarity matrix of five pairs of textual and visual components for relation and entity using CLIP-ViT-B/32 and our own method. Our method exhibits superior performance, as evident by the higher values along the diagonal of the matrix. In addition to the diagonal values, other matrix elements may indicate semantic relevance, and our alignment demonstrates better performance in matching these.

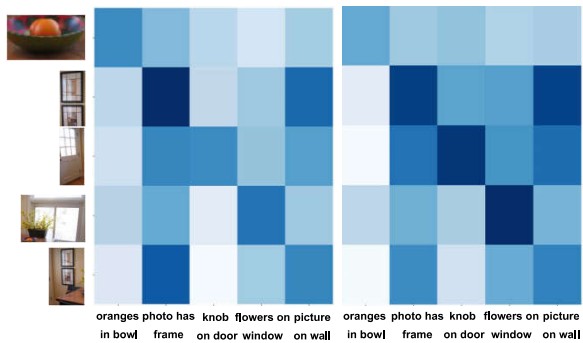 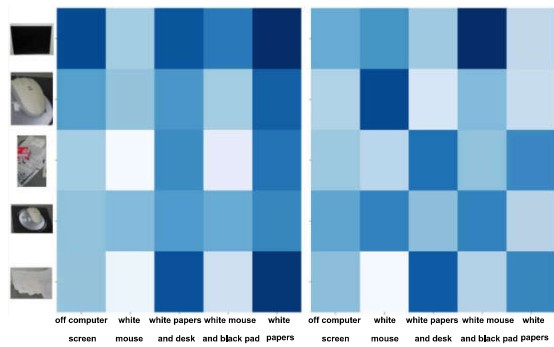

**Relational component similarity matrices**     **Entity component similarity matrices**

Figure 3: Illustration of relational and entity component similarity matrices. Left: CLIP-ViT-B/32, Right: ComAlign (Ours).

## 5    Conclusion

In this paper, we proposed an alignment model to enhance the compositional understanding of VLMs while maintaining the coarse-grained features. Our approach involves extracting fine-grained entity and relational components and proposing a strategy to match the corresponding components across modalities. We have shown that it is possible to align the base VLMs using a lightweight network and a relatively small dataset to utilize their fine-grained and compositional capacity more efficiently. By enhancing the fine-grained and compositional understanding of VLMs, we improve retrieval, compositional understanding, and downstream tasks.

**Limitations and Future Works** Although our method incorporates elements of text structure, it fails to comprehend the direction of relationships between objects. Furthermore, we do not fully utilize the entire graph structure; instead, we only match nodes and edges of relational components. Future works can involve addressing these limitations to potentially improve performance.

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

## A  Appendix

### A.1  Zero-shot Classification

In the following section, we conduct an evaluation of our proposed method on the zero-shot image classification task using five downstream classification datasets. Since models designed to enhance compositional understanding typically aim to improve retrieval performance on inputs involving multiple objects, they are not generally expected to boost performance on classification tasks, which often involve images containing only a single object. Consequently, in line with the findings reported in Yao et al. (2021); Gao et al. (2022), if these models do not degrade zero-shot classification performance, the results will be considered as satisfactory. Table 5 illustrates the results of our alignment method compared to the base VLMs across these datasets. The results of our zero-shot image classification experiments indicate that our method yields comparable performance to existing models across various datasets. While we observed improvements in certain cases, other results showed minimal changes or slight decreases. This discrepancy likely contributes to the mixed results observed in our zero-shot classification experiments.

To perform classification we create a template for each class and perform retrieval based on these templates. Similar to the inference process described in Section 3.4, we use a linear combination of the base VLMs' coarse-grained features and our coarse-grained embeddings to calculate the final similarity score.

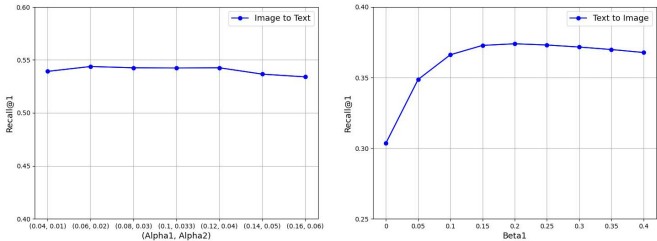

Figure 4: Impact of different values of $\alpha_1$, $\alpha_2$, and $\beta_1$ on I2T and T2I retrieval on our validation set.

Table 5: Top-1 accuracy of zero-shot image classification on 5 datasets.

| Method | Cifar10 | Cifar100 | Caltech101 | SUN397 | ImageNet |
|---|---|---|---|---|---|
| CLIP-ViT-B/32 | 89.82 | 63.85 | 82.53 | 46.41 | 57.57 |
| CLIP-ViT-B/32 + ComAlign | 89.82 | 64.35 | 82.68 | 47.02 | 57.92 |
| Relative Gain | 0.00 | 0.50 | 0.15 | 0.61 | 0.35 |
| CLIP-ViT-L/14 | 95.84 | 77.15 | 84.92 | 69.19 | 71.01 |
| CLIP-ViT-L/14 + ComAlign | 96.02 | 77.45 | 85.09 | 69.88 | 71.23 |
| Relative Gain | 0.18 | 0.30 | 0.17 | 0.69 | 0.22 |
| COCA-ViT-B/32 | 93.69 | 74.27 | 85.84 | 65.57 | 58.68 |
| COCA-ViT-B/32 + ComAlign | 93.97 | 73.88 | 85.64 | 65.19 | 58.22 |
| Relative Gain | 0.28 | -0.39 | -0.20 | -0.38 | -0.46 |
| NegClip-ViT-B/32 | 86.05 | 60.68 | 78.86 | 54.64 | 49.56 |
| NegClip-ViT-B/32 + ComAlign | 85.99 | 60.79 | 79.04 | 54.96 | 49.83 |
| Relative Gain | -0.06 | 0.11 | 0.18 | 0.32 | 0.27 |
| ViTamin-B-LTT | 90.71 | 78.19 | 87.60 | 62.13 | 66.91 |
| ViTamin-B-LTT + ComAlign | 95.57 | 78.36 | 87.22 | 62.50 | 67.25 |
| Relative Gain | 4.86 | 0.17 | -0.38 | 0.37 | 0.34 |
| MobileCLIP-B | 95.72 | 84.26 | 89.09 | 68.34 | 69.48 |
| MobileCLIP-B + ComAlign | 97.80 | 86.16 | 88.67 | 69.12 | 72.16 |
| Relative Gain | 2.08 | 1.90 | -0.42 | 0.88 | 2.68 |

## A.2  Hyper-parameter Tuning

Hyper-parameters of Equations 10 and 11 have been tuned utilizing a subset of MS-COCO Training split accomplished by ViT-B/32 model which trained on VisualGenome and Flickr30k datasets. Our experimented results are illustrated in Figures 4.

Also, we report the performance of our method under different hyper-parameters in zero-shot image-text retrieval on MSCOCO. Figure 5 shows the results of using various batch sizes, learning rates, and training epochs, as well as different coefficients for the coarse-grained contrastive loss.

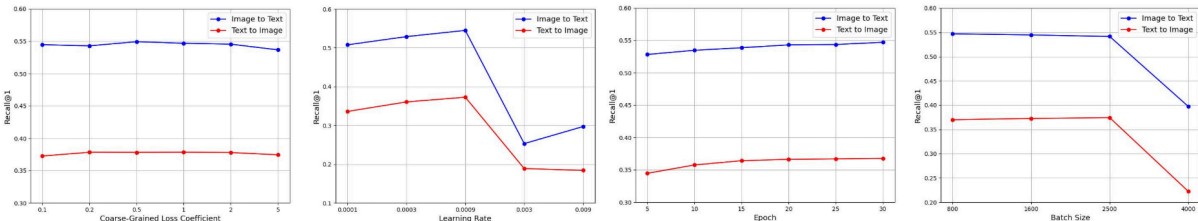

Figure 5: Experiments with different coefficients of coarse-grained contrastive loss, learning rate, number of epochs, and batch sizes on I2T and T2I zero-shot retrieval on MSCOCO.

