# OpenReview forum: "ComAlign: Compositional Alignment in Vision-Language Models"
_TMLR — Rejected by TMLR_

### Review · Reviewer_rjmD · 2025-06-07

**Summary Of Contributions:**

The paper introduces ComAlign, a method to enhance the compositional understanding of Vision-Language Models (VLMs) by aligning fine-grained components (entities and relations) across image and text modalities. The method tries to address the limitations of existing VLMs, which often fail to capture nuanced relationships and attribute bindings due to coarse-grained contrastive training.

**Audience:**

No

**Broader Impact Concerns:**

This work presents no apparent ethical risks or negative societal impacts.

**Claims And Evidence:**

No

**Requested Changes:**

The paper's contributions are weakened by insufficient theoretical grounding, a heuristic-driven relation modeling strategy, and limited novelty in methodology. Strengthening the discussion on why the method works and how it advances beyond existing techniques would significantly improve its academic rigor.

**Strengths And Weaknesses:**

#### **Strengths**
- **Clear Method Presentation**:
  The paper provides a well-structured and reproducible description of ComAlign, including specific implementation details (e.g., SpaCy's `en_core_web_sm` for text processing and YOLOv9 for visual component extraction). This clarity facilitates empirical validation and adoption.

#### **Weaknesses**
1. **Technique Report Style Over Theoretical Analysis**:
While the paper thoroughly documents technical details, it reads more like a technique report than an academic paper. The discussion on the motivation and underlying principles of the method is notably sparse. A deeper analysis of why the proposed alignment strategy works—beyond empirical improvements—would strengthen the contribution.

2. **Questionable Compositional Structure Modeling**:
The method claims to capture compositional relationships in images but relies on a heuristic scoring mechanism for visual relations (i.e., the product of entity confidence scores). This approach prioritizes high-confidence entity pairs rather than explicitly modeling their semantic or spatial relationships, undermining the premise of learning true compositional structure.

3. **Incremental Technical Contribution**:
  The work largely combines existing techniques (e.g., contrastive learning with local feature matching, FILIP’s fine-grained alignment strategy) without introducing significant conceptual innovation. While the integration is effective, the paper does not provide deeper theoretical insights or a clear advancement over prior methods. A more rigorous discussion of how ComAlign differs from or improves upon related approaches would better justify its novelty.

---

### Review · Reviewer_S85h · 2025-06-16

**Summary Of Contributions:**

This work presents ComAlign, a method to mitigate the limitations of VLMs in capturing compositional structure.
ComAlign introduces a new fine-tuning objective that uses entity and relation components from both the image and text modality.
For text, it uses a pre-trained SpaCy language model to derive entity and relation triplets.
For images, it identifies visual entities (object bounding boxes) and their spatial relationships.

This additional entity information is then integrated as input to the frozen image and text encoders.
Furthermore, two lightweight transformer networks are trained to extract this fine-grained entity information. The core of ComAlign's learning lies in its use of two fine-grained matching losses to align image and text entities and relations, alongside the conventional global contrastive loss.
The effectiveness of ComAlign is supported by extensive zero-shot image-text retrieval results on benchmarks such as MS-COCO, Flickr30K, VG-Att, VG-Rel, and SVO-Probes, showcasing its improved performance across diverse backbones.

**Audience:**

Yes

**Broader Impact Concerns:**

-

**Claims And Evidence:**

No

**Requested Changes:**

- Can the authors argue why they freeze the encoders instead of tuning the top-k layers? Can an experiment be added where the encoders are trained by unfreezing the last two layers, without adding extra transformer layers?
- Can a figure be added to the introduction to provide a high-level overview of the method? This will make it easier to understand what the proposed method does at the start of the paper.
- Can the authors use \citep for inline citing?
- Can the authors add experiments that clearly show what the gain is of the FGM losses, because they seem very marginal? Also, can some variance estimates be added?
- Can the authors explain why global loss + {entity, relation} is worse than global loss alone?
- Can the claim: "Our method assumes that VLMs like CLIP can extract initial representations for entities of the text and objects of the image and need to be empowered by a lightweight model that can align the structure of the visual and textual modalities." be justified?
- Can a model be trained from scratch (i.e., VSE++ or something) using the new loss function and compare it to contrastive only?

**Strengths And Weaknesses:**

Strength:
- A new optimization objective is introduced for VLMs.
- Extensive results to show the effect of the newly introduced loss terms/model inputs.

Weakness:
- In Table 3, an ablation is provided for each loss term.
However, the numbers using the three losses together do not match those in Table 1 for VIT-B/32, since they are run on the validation set. This discrepancy also indicates variance in the obtained results, and no confidence intervals are given for any of the numbers in the paper. Moreover, this table suggests that training with the global loss alone already accounts for the majority of the observed gain, which is akin to simply fine-tuning with some additional transformer layers. Furthermore, training with global loss + {entity, relation} even results in lower I2T outcomes than global loss alone, with only a marginal improvement for T2I, which is strange. In my opinion, the observed results can mainly be explained by adding more parameters to the model and incorporating additional dimensions to the training data that the model can effectively utilize. The increased performance can hardly be attributed to a better compositional understanding of the model.
- "Our method assumes that VLMs like CLIP can extract initial representations for entities of the text and objects of the image and need to be empowered by a lightweight model that can align the structure of the visual and textual modalities." -> Where is this assumption based on? I find this a very strong assumption without any evidence. Even more so since the main argument of the paper is that VLMs cannot represent fine-grained compositional information.
- (related to the point above) I find it very intuitive that the main argument of the paper is "VLMs struggle to represent binding
attributes to the corresponding objects or identifying relationships between subjects and objects". If the frozen encoder doesn't represent these features in the first place, then simply adding more layers on top won't extract what isn't there. It is just overfitting.

---

### Review · Reviewer_aRoM · 2025-06-20

**Summary Of Contributions:**

This work introduces ComAlign, a lightweight and efficient method for improving the compositional understanding of Vision-Language Models (VLMs). Addressing the weak compositional structure in models trained with coarse-grained loss, ComAlign trains a small plugin network on top of frozen VLM encoders using a minimal dataset of image-text pairs. The method's key contribution is a fine-grained approach to discover more exact correspondence of text and image pairs using the weak supervision in image-text pairs.. By explicitly aligning these fine-grained components across modalities using a specialized matching strategy, ComAlign enhances the performance of various base models on compositional and retrieval benchmarks, without needing to retrain the entire VLM.

**Audience:**

Yes

**Claims And Evidence:**

No

**Requested Changes:**

The authors are encouraged to highlight the technical contributions and novelities compared to previous works on fine-grained alignment and this work's advantages.

**Strengths And Weaknesses:**

Strengths:

1: Efficient and Practical Plug-in Architecture. The paper introduces ComAlign as a lightweight network that is trained on top of existing, frozen vision-language models (VLMs). This parameter-efficient approach avoids the need for retraining massive base models from scratch, which is computationally expensive. The authors demonstrate that this small, two-layer transformer network can be trained effectively on a relatively small dataset (~100k image-text pairs) in under four hours on a single GPU.

2: Performance Improvements. The paper reports substantial performance gains across multiple benchmarks and base models. For instance, ComAlign improves the I2T retrieval R@1 score of CLIP-ViT-B/32 on MSCOCO by 5.60% and T2I by 6.27%. Notably, the enhanced models often outperform PyramidCLIP, a model trained from scratch on a massive 143M sample dataset, despite ComAlign's much greater efficiency.

3: An automated date processing pipeline: The method introduces a preprocessing pipeline to explicitly extract and align meaningful compositional components—specifically entities and relations—from both image and text modalities using only weak supervision. Unlike prior models that perform coarse-grained alignment or treat image patches and text tokens as independent components, ComAlign ensures that the compositional structure extracted from text is preserved and matched in the visual modality.

Weaknesses:

1: The technical contributions are limited. The paper's core technical contribution, while effective, appears somewhat incremental. The authors argue that the coarse, global-level alignment often overlooks the compositional structure of images and text. Their proposal is to create a more fine-grained alignment by extracting and matching potential entity and relational components from both modalities. However, the foundational idea of moving beyond global alignment to leverage more granular components is not new. Similar concepts have been central to prior work such as FILIP [1], DetCLIP [2], and GLIP [3]. While the paper does contrast its method with FILIP's patch-and-token approach, it would significantly strengthen the work to provide a clearer, more direct comparison against other fine-grained alignment strategies.

2: In-depth analysis of the data preprocessing pipeline. One of the most important parts of the paper is the proposed pipeline for generating entities and relations, given only an overall text description. What will be the effect if we ablate some steps inside this pipeline? For example, would the adjective words be important for zero-shot understanding? Would there be a more complex relation type that needs considering, like, more than two entities? Does the bottleneck exist in the object detector? (I just list some random ideas.) Overall, I mainly hold concerns about the qualities of the constructed data pairs and the reasonableness of the match pairs.

3: An open discussion. From my perspective, this paper focuses more on building relations. Would the proposed method be better at open-vocabulary HOI detection or some other tasks beyond zero-shot classification and retrieval?

[1] Filip: Fine-grained interactive language-image pre-training

[2] Detclip: Dictionary-enriched visual-concept paralleled pre-training for open-world detection

[3] Grounded language-image pre-training

---

### Decision · Action_Editor_m9Gh · 2025-07-19

**Recommendation:** Reject

**Audience:**

Yes

**Audience Explanation:**

The topic of how to improve CLIP model is relevant to the TMLR's audience.

**Claims And Evidence:**

No

**Claims Explanation:**

This paper introduces ComAlign, a method to enhance the compositional understanding of Vision-Language Models (VLMs) by aligning fine-grained components (entities and relations) across image and text modalities. The paper received 2 Reject and 1 Leaning Reject recommendations.

Several major concerns exist. (1) The proposed method needs a more in-depth analysis of the pipeline on how to generate fine-grained components for alignment. (2) Experiments and analysis are not convincing enough, as noted in the reviewers' comments. No rebuttal was provided; therefore, the concerns remain unaddressed. The AC would like to recommend rejection of the paper.